



# Characterisation of fault plane and coseismic slip for the May 2, 2020, Mw 6.6 Cretan Passage earthquake from tide-gauge tsunami data and moment tensor solutions

Enrico Baglione[1,2], Stefano Lorito[2], Alessio Piatanesi[2], Fabrizio Romano[2], Roberto Basili[2], Beatriz Brizuela[2], Roberto Tonini[2], Manuela Volpe[2], Hafize Basak Bayraktar[3,2], Alessandro Amato[2]

[1]Istituto Nazionale di Oceanografia e di Geofisica Sperimentale (OGS)- Sgonico (TS) – Italy
[2]Istituto Nazionale di Geofisica e Vulcanologia, Sezione di Roma 1, Via di Vigna Murata 605, 00143, Roma, Italy
[3]Department of Physics "Ettore Pancini", University of Naples Federico II, Naples, 80126, Italy

*Correspondence to*: Enrico Baglione (enrico.baglione@ingv.it)

**Abstract.** We present a source solution for the tsunami generated by the Mw 6.6 earthquake that occurred on May 2, 2020, about 80 km offshore south of Crete, in the Cretan Passage, on the shallow portion of the Hellenic Arc Subduction Zone (HASZ). The tide-gauges recorded this local tsunami on the southern coast of Crete island and Kasos island. We used these tsunami observations to constrain the geometry and orientation of the causative fault, the rupture mechanism and the slip amount. We first modelled an ensemble of synthetic tsunami waveforms at the tide-gauge locations, produced for a range of earthquake parameter values as constrained by some of the available moment tensor solutions. We allow for both a splay and a back-thrust fault, corresponding to the two nodal planes of the moment tensor solution. We then measured the misfit between the synthetic and the observed marigrams for each source parameter set. Our results identify the shallow steeply-dipping back-thrust fault as the one producing the lowest misfit to the tsunami data. However, a rupture on a lower angle fault, possibly a splay fault, with a sinistral component due to the oblique convergence on this segment of the HASZ, cannot be completely ruled out. This earthquake reminds us that the uncertainty regarding potential earthquake mechanisms at a specific location remains quite significant. In this case, for example, it is not possible to anticipate if the next event will be one occurring on the subduction interface, on a splay fault, or on a back-thrust which seems the most likely for the event under investigation. This circumstance bears important consequences because back-thrust and splay faults might enhance the tsunamigenic potential with respect to the subduction interface due to their steeper dip. Then, these results are relevant for tsunami forecasting both in the framework of the long-term hazard assessment and of the early warning systems.



## 1 Introduction

On May 2, 2020, at 12:51:07 UTC, a strong earthquake occurred in the Cretan Passage, about 80 km offshore to the south of Crete Island in the eastern Mediterranean. According to the revised moment tensor solution distributed by the GEOFON (https://geofon.gfz-potsdam.de/), the earthquake was located at 25.75°E and 34.27°N, at a depth of 10 km, and the moment magnitude (Mw) was 6.6 (Figure 1). Within about 10-15 minutes after the event, estimates of the earthquake magnitude varied from Mw 6.5 to 6.7. This appears, for example, from tsunami alerts issued by the three Tsunami Service Providers (TSPs) of the Tsunami Early Warning and Mitigation System in the North-eastern Atlantic, the Mediterranean and connected seas (NEAMTWS, http://www.ioc-tsunami.org/), in charge for monitoring this region: the Centro Allerta Tsunami - Istituto Nazionale di Geofisica e Vulcanologia (CAT-INGV), the National Observatory of Athens (NOA), and the Kandilli Observatory and Earthquake Research Institute (KOERI). These estimates were then confirmed by the moment tensor solutions which started to appear immediately after (Figure 2a).

The 2020 Cretan Passage earthquake generated a local tsunami along the south-eastern coast of Crete, as reported by eyewitnesses and local authorities and documented by a series of pictures and video shootings taken by authorities, press, and amateurs at Arvi and Kastri villages (Papadopoulos et al., 2020). The NOA-04 tide-gauge station, located in the port of Ierapetra, recorded a peak-to-trough excursion exceeding 30 cm, with a positive peak amplitude of about 20 cm recorded 23 minutes after the earthquake origin time, with a wave period of ~3.5 minutes. Small tsunami waves (less than 10 cm peak-to-trough) were also recorded at the NOA-03 tide-gauge, located in the Kasos Island, where the peak amplitude of 5 cm was recorded at 13:53 UTC, and the wave period was estimated to be 8 minutes by Papadopoulos et al. (2020) and 4.5 minutes by Heidarzadeh and Gusman (2021). As in the Mw 6.4, July 1, 2009, event (Bocchini et al., 2020), the tsunami was also observed in the Chrysi islet (located offshore south of Ierapetra), where no tide-gauges are operating. No casualties, injuries or damage were reported due to the tsunami.

The 2020 Cretan Passage earthquake occurred in the Hellenic Arc Subduction Zone (HASZ). The HASZ is the active plate boundary that accommodates the convergence of the African (or Nubia) plate sinking under the Aegean plate. The arc stretches NW-SE from Kefalonia-Lefkada to Crete and SW-NE from Crete to Rhodes. According to GPS velocities, the relative motion across the HASZ is ~30 mm/y in the NE-SW direction (Nocquet, 2012). The HASZ is characterised by an active volcanic arc in the southern Aegean Sea, an outer non-volcanic arc marking the transition from back-arc extension to contraction in the forearc along the Ionian Islands, Crete, and Rhodes (backstop), a complex accretionary wedge characterised by alternating forearc basins, known as part of the Hellenic Trench (or Trough) System (Matapan, Poseidon, Pliny, and Strabo basins, Fig. 1) and Inner Ridges, and the more external, thicker, and wider, Mediterranean Ridge. The accretionary wedge extends above the oceanic crust for more than 200 km, with its leading-edge affecting the remaining abyssal plains (Ionian, Sirte, and Herodotus) and nearing the African continental margin (Polonia et al., 2002; Kopf et al., 2003; Chamot-Rooke et al., 2005;


Yem et al., 2011) and has an outward growth rate of 5-20 mm/y (Kastens, 1991). According to reconstructions based on seismic
reflection data, most of the structural characteristics of the Mediterranean Ridge external domain can be explained by the
presence of thick Messinian evaporites, whereas the internal structures include both frontal thrusts and back-thrusts
(Chaumillon and Mascle, 1997; Kopf et al., 2003). Back-thrusts mainly characterise the transition of the Mediterranean Ridge
to the inner domain. Strike-slip motions are also present within the Hellenic Trench system.

Several strong earthquakes struck this area in the past. The largest documented earthquake is the Mw~8.3 365 CE event that
occurred in the central forearc of the subduction zone southwest of Crete (Papazachos et al., 2000; Papazachos and Papazachos,
2000; Stiros, 2001). This earthquake generated a devastating tsunami (Guidoboni and Comastri, 1997). Another remarkable
event is the Mw~8 earthquake of August 8, 1303, which occurred southeast of Crete island, specifically in the arc portion
between Crete and Rhodes (Guidoboni and Comastri, 1997, Papazachos, 1996). This earthquake was probably the cause of a
tsunami that affected Alexandria in Egypt (Guidoboni and Comastri, 1997). Other strong tsunamigenic earthquakes in the
easternmost Hellenic Arc are the Mw 7.5, May 3, 1481 event (Yolsal-Çevikbilen and Taymaz, 2012) and the Mw 7.5, January
31, 1741 (Papadopoulos et al., 2007) one. The occurrence of the 1303, 1481 and 1741 tsunamis is also geologically testified
by sediments found on the Dalaman coast (Papadopoulos et al., 2014). Another large tsunamigenic earthquake (M ~ 7.0–7.5)
occurred near southern Crete on July 1, 1494 (Yolsal-Çevikbilen and Taymaz, 2012). More recently, an earthquake of Mw 7.5
occurred on February 9, 1948, near the coast of Karpathos, on the Pliny Trench (Papadopoulos et al., 2007) and, on July 1,
2009 (UTC 09:30), a moderate earthquake (Mw 6.5) located in the southern offshore margin of Crete caused a local tsunami
of about 0.3 m of wave height (Bocchini et al., 2020).

Despite the relatively high seismicity documented by decades of investigations in macroseismic and instrumental historical
seismology in the eastern Mediterranean, several aspects of the tectonic and geodynamic processes that characterise the
Hellenic forearc deserve further investigations. For example, the transition from extension to contraction in the forearc is not
well delimited, and even the type of seismogenic activity at the subduction interface is not entirely clear.

For example, the great 365 CE earthquake has been associated with different crustal faults in the upper plate: a reverse splay
fault (Shaw et al., 2008; Shaw and Jackson, 2010; Saltogianni et al., 2020) and, recently, a pair of orthogonal normal faults
(Ott et al., 2021). Conversely, it seems that the 1303 event was due to a rupture on the plate interface itself (Papadopoulos,
2011; Saltogianni et al., 2020). Two recent earthquakes that occurred near the 2020 Cretan Passage event were attributed to
two different mechanisms. The source of the recent Mw 6.5, July 1, 2009, earthquake that triggered a small tsunami was
suggested to be a splay fault (Bocchini et al., 2020). The Mw 5.5, March 28, 2008, earthquake that occurred to the south of
Crete was instead attributed to a north-dipping low-angle thrust faulting mechanism with a small amount of left-lateral slip
component (Shaw and Jackson, 2010; Yolsal-Çevikbilen and Taymaz, 2012) representing the subduction interface.



Although all the envisaged mechanisms of these examples are consistent with the variety of mechanisms that characterise a
subduction zone, the study of the seismogenic and tsunamigenic sources south of Crete remains of key importance for
improving the characterisation of the associated hazards, which affects the nearby inhabited coastal areas. This region was
already identified as subject to relatively high seismic and tsunami hazard (e.g., Sørensen et al., 2012; Woessner et al., 2015;
Basili et al., 2021), and a better characterisation of the potential sources may reduce the uncertainty of such estimates.
Other authors have already studied the 2020 Cretan Passage event. In particular, Heidarzadeh and Gusman (2021) studied the
tsunami source and obtained a heterogenous slip model by inversion and spectral analysis of the tsunami records. They impose
a fixed fault geometry for their model, that is one of the two nodal planes (strike, 257°; dip, 24°; rake 71°) of the GCMT
solution (Dziewonski et al., 1981; Ekström et al., 2012). This solution is a north-dipping plane compatible with a dominantly
thrusting mechanism on a splay fault. The fault centre is placed roughly in the middle between the United States Geological
Survey (USGS) epicentre (25.712° E, 34.205° N) and the GCMT centroid location (25.63° E, 34.06° N).
Here, we invert tsunami data for the fault location and orientation (strike and dip angles) as well as for the earthquake-average
slip amount and direction (rake angle). To limit the solutions to be explored, we first constrain the parameters to range around
the values of the available moment tensor solutions. In this way, while focusing on solutions compatible with the moment
tensor inversions of seismic data, we do not exclude a priori that the earthquake might have happened on either nodal planes
of these mechanisms. Then, we produce the synthetic tsunami waveforms at the Ierapetra and Kasos tide-gauges for all the
sources we obtained. Lastly, we calculate the misfit with observed signals, analyse the misfit distribution for the whole
ensemble of models explored, and derive the most likely source model for this earthquake.



**Figure 1:** Main seismotectonic elements of the Hellenic Arc Subduction Zone (HASZ). The seismicity is derived by the SHEEC-EMEC (Grünthal and Wahlström, 2012; Stucchi et al., 2013) and NOA (http://www.gein.noa.gr/en/seismicity/earthquake-catalogs) earthquake catalogues. Focal mechanisms are from the Global Centroid Moment Tensors database (GCMT; Dziewonski et al., 1981; Ekström et al., 2012). The slab depth contours are resampled from the European database of Seismogenic Faults (EDSF) (Basili et al., 2013). The topo-bathymetry is obtained by splicing the ETOPO1 Global Relief Model and EMODnet Digital Bathymetry (DTM 2020) (NOAA, 2009; Amante and Eakins, 2009; EMODnet Bathymetry Consortium, 2020). The black rectangle outlines the area shown in Figure 2a.



## 2 Data and Methodology

We compared the sea level observations at the two tide-gauges (Ierapetra and Kasos) with the synthetic waveforms obtained through numerical tsunami simulations, to identify the source that produced the tsunami based on many different sets of fault parameters. In this section, we describe the technical details of our approach.

### 2.1 Seismic source parameterization

We tested different combinations of source parameters, considering 41,310 solutions (Table 1). Each solution is represented by a rectangular fault with uniform slip. The length and width of the fault were assigned based on fault scaling relationships (Leonard, 2014) for a fixed moment magnitude Mw = 6.6. We varied position, depth, strike, dip, rake, and slip.

The earthquake struck in a region where hypocentral locations are usually poorly constrained (Bocchini et al., 2020). The use of a different number of seismic stations, the type of phases used (namely at local, regional or teleseismic distances) and the choice of velocity models can lead to a significant discrepancy in hypocentral locations. The centre of the rectangular fault is thus allowed to span different values of latitude, longitude, and depth (Table 1) to consider this variability.

Strike, dip, and rake are explored by regular steps within a range of values that envelopes the focal mechanism solutions provided by several agencies (GFZ, USGS, GCMT, IPGP; Figure 2a). Two classes of nodal planes are explored; one is a north shallow-dipping plane, coherently with the dip direction of the subduction interface in that region, or a splay fault (hereafter called "plane S"), the other one is a steep south-dipping plane, likely identifying a back-thrust ("plane B"). Some "extreme" values, like a dip larger than 70° for plane B or lower than 20 for plane S, have been excluded after some preliminary tests, as they were significantly worsening the misfit between synthetic and observed waveforms. Slip is allowed to vary between 0.35 and 1.15 m, with a step of 0.05 m.

**Table 1:** Source parameters variability of the source model dataset for the tsunami simulations. The different sets of focal plane parameters are separated by parenthesis (B and S refer to the back-thrust and splay fault solutions). Positions and depths are referred to the centre of the fault plane.

| Source parameters | |
|---|---|
| Length (km) | 26.04 |
| Width (km) | 15.42 |
| Depth (km) | 10; 15; 20 |
| Lat (°N) | 34.1; 34.2; 34.3 |
| Lon (°E) | 25.6; 25.7; 25.8 |
| Slip (m) | from 0.35 to 1.15, step 0.05 |
| Strike (°) | B (95; 105), S (225; 235; 245; 255; 265) |




| Dip (°) | B (50; 60; 70), S (20; 30; 40) |
|---|---|
| Rake (°) | B (85, 95; 105; 115, 125), S (45; 55; 65; 75) |

147

148

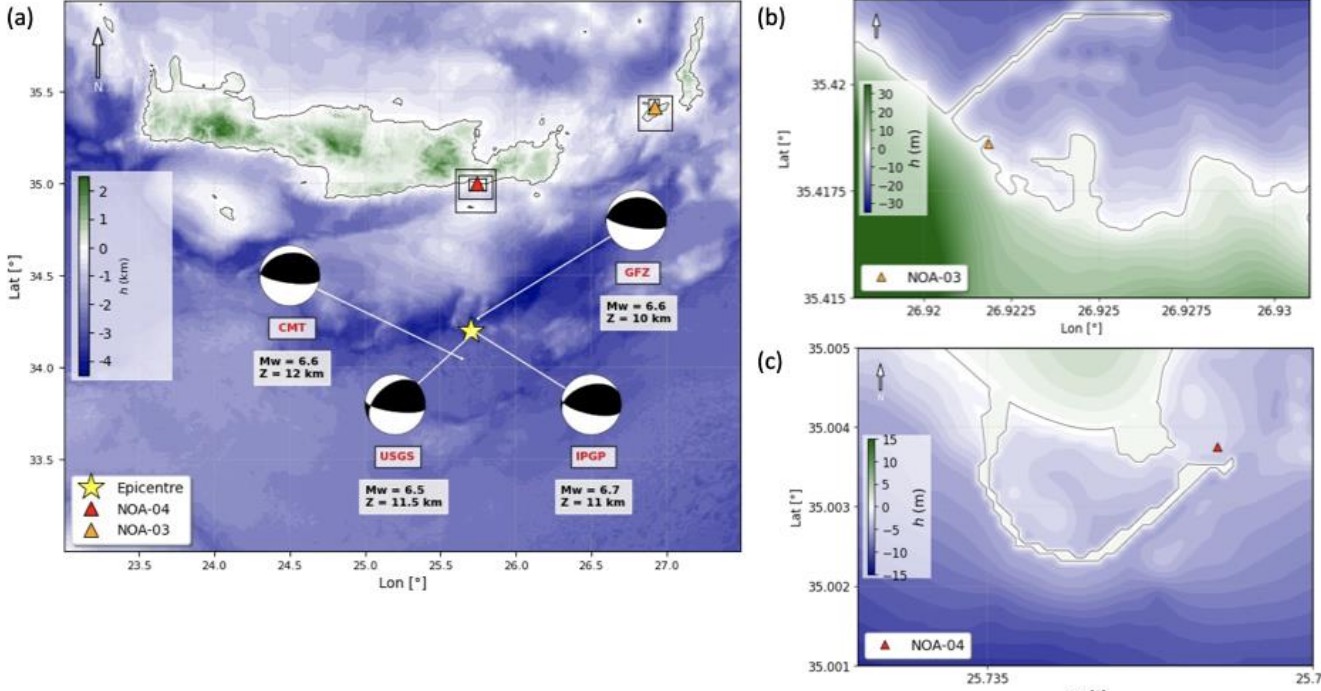

149

**Figure 2:** (a) Computational domain for the tsunami modeling adopted in this study (see text for details). The yellow star indicates the epicentre, at the centre (34.2°N, 25.7°E) of its considered variability range. THe different bathymetric levels are plotted as black rectangles. The red and orange triangles represent the Ierapetra (NOA-04) and Kasos (NOA-03) tide-gauge stations, respectively. The different focal mechanisms used as reference values to let the inversion parameters vary are plotted, each with its own agency label: GEOFON (GFZ, https://geofon.gfz-potsdam.de/), United States Geological Survey (USGS, https://earthquake.usgs.gov/), Institut de Physique du Globe de Paris (http://geoscope.ipgp.fr/), Global CMT Catalog (https://www.globalcmt.org/). (b) High-resolution bathymetry data (10 m spatial resolution) around NOA-03 (Kasos) and (c) NOA-04 (Ierapetra) tide-gauges.

### 2.2 Tide-gauge data and tsunami modelling

The tsunami signal recorded by the tide-gauges at Ierapetra (NOA-04) and Kasos (NOA-03) was obtained after removing the tidal component from the original waveform (http://www.ioc-sealevelmonitoring.org, sampling rate of 1 min) through a LOWESS procedure (e.g., Romano et al., 2015).

161



Tsunami numerical modelling was performed with the Tsunami-HySEA software, which uses a finite volumes approach and a nested grid scheme to progressively increase the resolution during the propagation from the source to the tide-gauges. The software has undergone proper benchmarking (Macías et al., 2017) according to the community standards (e.g., Synolakis et al., 2009), also within the framework of the US tsunami hazard program (http://nws.weather.gov/nthmp/). The code is implemented in CUDA (Compute Unified Device Architecture) and runs in multi-GPU architectures, yielding remarkable speedups compared to other CPU-based codes (de la Asunción et al., 2013).

To build the bathymetric and topographic grid models for the simulations, we used: 1) the European Marine Observation and Data Network (EMODnet) project database (EMODnet DTM version released in 2018, http://portal.emodnet-bathymetry.eu/), which has a resolution of about 115 m; 2) the European Digital Elevation Model (EU-DEM), version 1.1 (eu_dem_v11_E50N10), with a resolution of 25 m; and 3) the nautical charts (https://hartis.org/en) of Ierapetra harbour (Ierapetra Bay, 1:10,000 scale; Kaloi Limenes Bay, 1:12,500 scale) and Kasos harbour (Diafani Harbour, 1:5,000 scale; Pigadia Bay and Harbour, 1:5,000 scale; Emporio Harbours, 1:5,000 scale). The computational domain (33-36° N, 23-27.5° E, Figure 2a) for tsunami propagation consisted in four levels of nested grids with increasing resolution approaching the Ierapetra and Kasos harbours (640, 160, 40, and 10 m, respectively). The domains of the finest grids are shown in Figures 2b and 2c.

The instantaneous seafloor vertical displacement was calculated using Volterra's formulation of elastic dislocation theory applied to a rectangular source embedded in an elastic half-space (Okada, 1992), and the initial velocity field is assumed to be zero everywhere. The initial sea surface elevation was obtained by applying a low-pass filter to reproduce the water column attenuation; the filter has a trend of the type $1/\cosh(kh)$, where "k" is the wavenumber, and "h" is the average water depth (Kajiura, 1963).

We performed 2,430 simulations exploring all the source parameters (Table 1) except for the slip, which is fixed in all runs to 1 meter to obtain Green's functions. For all of these scenarios, we simulated one hour of propagation after the earthquake origin time (hereinafter OT) for the Ierapetra station and one hour and 30 minutes of propagation for the Kasos station. These simulation lengths allowed us to have about 50 minutes of tsunami signal at both gauges, which is more than enough to include the first tsunami oscillations (~30 min), that carry the information on the source and are used for the inversion (see Section 2.3). Time histories of the tsunami waves were calculated at the wet points of the computational grid closest to the Ierapetra and Kasos station coordinates (see Figure 2) and stored every 60 s, consistently with the actual tide-gauge sampling. We assumed linearity between the slip amount and the tsunami to obtain the scenarios for different slip values. Thus, we multiplied each of the computed marigrams by all the 17 slip values, for a total of 41,310 tsunami realisations.





**2.3 Inversion**

To retrieve the fault parameters and the coseismic slip simultaneously, we solved a nonlinear inverse problem. Since the number of sources in our ensemble is not very large, we opted for a systematic search of the parameters' space.

The comparison between the synthetic and the observed waveforms is carried out in the time domain. The misfit between the two waveforms is evaluated through a cost function frequently used to compare tsunami signals in source inversions (e.g., Romano et al., 2020):

$$E = 1 - \frac{\sum_{t_i}^{t_f} \eta(t-T)\eta_0(t)}{\sum_{t_i}^{t_f} \eta^2(t-T) + \sum_{t_i}^{t_f} \eta_0^2(t)},$$ (1)

In equation (1) $\eta(t)$ and $\eta_0(t)$ are the synthetic and the observed waveforms, respectively, $t_i$ and $t_f$ are the lower and upper limit of the considered time window, and $T$ is a time shift. The cost function considers both the amplitude and the shape of a waveform; it is more robust than a least-squares misfit, whose solutions are very sensitive to a small number of large errors in the dataset (Tarantola, 1987). For each combination of the source parameters, the cost function is minimised with respect to time shift values between -5 and 5 minutes, with one-minute steps. The arrival time optimisation is used to overcome the often found time alignment mismatch between the observed and modelled tsunami waveforms, with the latter generally arriving earlier. This approach was introduced by Romano et al. (2016), and the details are discussed further in Romano et al. (2020).

The overall cost function is a weighted average of two cost functions calculated on the two considered tide-gauges. The weights are chosen such that $\frac{w_{NOA-03}}{w_{NOA-04}} = 0.2$, equivalent to the ratio of the maximum tsunami amplitude registered at the two tide-gauges in the first half an hour after the tsunami arrival. Several attempts were made, showing that the results are driven by the Ierapetra contribution for a wide range of weights. The higher sensitivity of the Ierapetra signal to the source details is not surprising, since the Kasos station is much further away from the source, and the associated recorded marigram shows a very low peak-to-trough excursion and a lower signal to noise ratio.

Time windows of [5, 30] and [30, 55] minutes after the earthquake OT are chosen, respectively, for the Ierapetra and Kasos tide-gauges. This choice was made to include the first tsunami oscillations, which are mainly driven by the seismic source. The remaining part of the records is not used for the inversion, because it is highly probable that other factors, such as the local propagation and the port structure, start to control the shape of the signal (Romano et al., 2016; Cirella et al., 2020). To quantify the relative importance of these factors, the cost function is also evaluated in the 25 minutes following the considered intervals, that is in the time windows [30,55] and [55,80] minutes for Ierapetra and Kasos, respectively. The average of the cost functions ($E_1$ for [30,55] min., $E_2$ for [55,80] min.) is calculated from the 5, 10, 50, and 100 percent of models with the lowest misfit $E_1$ (within the first window used for the inversion) with the observed data. We observe that the ratio $E_2/E_1$ significantly decreases


when using progressively more models ($E_2/E_1$ = 4.9, 4.5, 3.0, 2.0, respectively). This observation confirms that the information
about the source dominates the first intervals used for the inversion.

**2.4 Synthetic test**

We first investigated the resolution offered by the two stations using as a target source model all possible combinations of the
source parameters $A(a_1, a_1, …, a_n)$. These are the same models we explored in the inversion for the real case. For each of them
we calculated the corresponding synthetic target waveform and corrupted it by adding a Gaussian random noise with a variance
corresponding to the 10% of the clean waveform amplitude variance. A random time shift between -5 and 5 minutes is added
to mimic the typically observed time mismatch between the observed and the predicted tsunami signals.

All the waveforms $f(A)$ derived from all the possible source models are tested against each of these noisy and shifted target
waveforms $f_T(A)$ using equation (1). We then defined the distance between two different models as:
$$d_{ij} = \frac{\|a_i - a_j\|}{M \cdot \|a_j\|},$$   (2)
Where $a_i$= (strike, dip, rake, slip, depth, lon, lat)$_i$, $a_j$= (strike, dip, rake, slip, depth, lon, lat)$_j$ are the parameters associated with
the i-th (j-th) combination, and M (equal 7) is the number of free parameters.

For each target model $a_i$, the distance d is evaluated with respect to:

   1)   the best model $a_{best}$, whose $f(a_{best})$ presents the lowest cost function;

   2)   the average model $a_{wm}$ evaluated as a weighted mean over the first 5% of the models with the lowest cost function,
       where the weights are chosen as the reciprocal of the cost function.


The result confirms that the tsunami data well constrain the seismic source process. In most cases, the target parameters
correspond to those of the model which minimises the cost function (Figures 3a and 3c). Hence, the target focal plane is
correctly identified. The few cases showing a high value of the distance occurs when the algorithm does not recognise if the
target is a back-thrust or a splay fault.

On the one hand, when using the average model, the distance between the models almost never vanishes (Figures 3b and 3d),
meaning that the target's parameters are not perfectly reproduced, as expected for an average model. On the other hand, the
averaging process has the power to make the distribution smoother and unimodal and to eliminate or diminish the number of
occurrences corresponding to a high distance. So, choosing the average over the best models may protect us from overfitting.
Figure 3e shows that the B plane (a back-thrust) is much better spotted than the S one (the splay) by the best models; when
using the average model, the difference in the "specificity" of the cost function is slightly reduced but still present (Figure 3f).


**Figure 3:** Distributions of the parameters distance for the best (a, c, e subplots) and average models (b, d, f subplots). Subplots (a) and (b)

separate the models for which the target model focal mechanism is reproduced or not. Subplots (c) and (d) report all the models together.
Subplots (e) and (f) separate the target models associated with the B (red) or S (blue) focal plane solutions.
**3 Results of the application to the May 2, 2020, Mw 6.6 Cretan Passage earthquake**
We performed the inversion using the observations at Ierapetra and Kasos, the only two sea level recordings available. The
distribution of the cost function values for all the investigated models is shown in Figure 4. Figure 4a displays separately the
cost function values obtained for the two focal solutions. Overall, the cost functions of the B plane are slightly lower than those



of the S plane. However, the left portions of the distributions, that is the ones containing the models with the lowest misfit with
respect to the observed marigrams, are almost overlapped. The same tendency can be seen in Figure 4b where the distribution
has a slightly bimodal character with the two modes corresponding to the S and B planes, respectively.

Based on the resolution test results presented in Section 2, we evaluated the weighted average of the models included in the
5[th] percentile of the cost function distribution for each focal solution (those to the left of the dashed lines in Figure 4a). We
used as a weight the inverse of the cost function. Both the best and average models, as well as the associated errors obtained
as weighted standard deviations, are reported in Table 2.

The average models, along with the associated errors, may indicate that the best model is ''overfitting'' the data. This happens,
for example, when the best and average models are very different or when the uncertainties are very large. Standard deviations
give a measure of the uncertainties in the estimation of the corresponding parameter. Smaller values of the standard deviation
denote a parameters' better resolution (Mosegaard and Tarantola, 1995; Sambridge and Mosegaard, 2002; Piatanesi and Lorito,

279  2007).




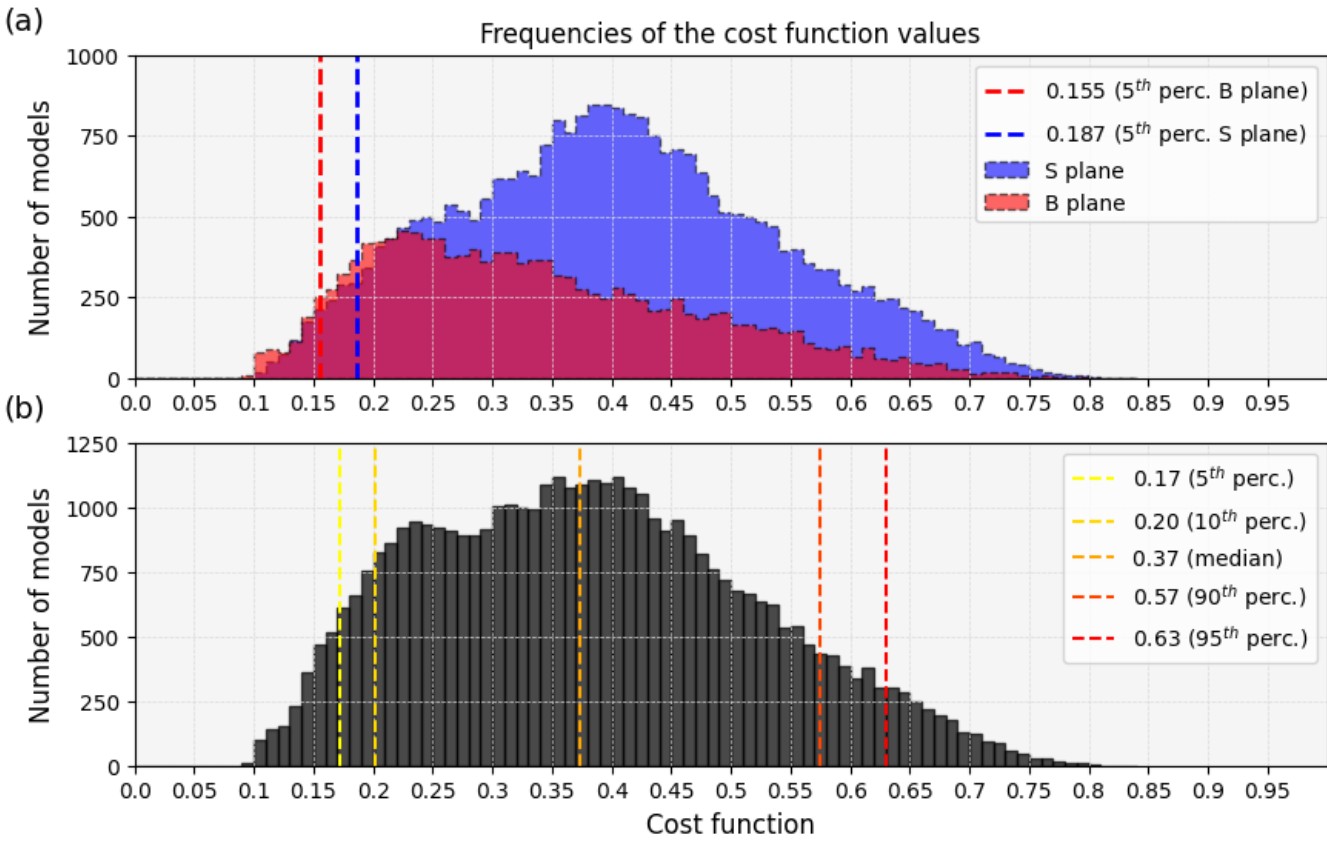

**Figure 4:** (a) Cost function distribution for the back-thrust (red) and the splay (blue) models; the vertical dashed lines indicate the 5th percentiles for each of the two focal solutions. (b) Histogram of the cost function values for all the models considered. The vertical dashed lines represent the 5th, 10th, 50th (median), 90th and 95th percentile.

With only a few exceptions, all the best model parameters fall within the range of one standard deviation from the average model. For both focal solutions, the slip of the best models is quite smaller than the average one and does not fall within the uncertainty limits.

The S plane solutions are centred about 10 km north of the B planes, slightly closer to the southern coast of Crete. Coherently, the predicted tsunami arrives earlier (i.e., the estimated time-shift is bigger) with respect to the waves resulting from the B plane solutions. The rake angle, both for B and S planes, presents a large dispersion. The same can be said for the strike associated with the S plane. On the other hand, the dip appears to be better constrained.




**Table 2:** Best and Average Model extracted from the models with the smallest cost functions within the 5[th] percentile. The percentiles
refer to B and S planes separately (i.e., the models at the left of the red and blue vertical dashed lines in Figure 4a, respectively). B plane
refers to the back-thrust solution dipping south; S plane refers to the splay fault dipping north. Lat, Lon and Depth refer to the centre of the
fault.

|  | Best model plane B | Average model (5[th]) plane B | Best model plane S | Average model (5[th]) plane S |
|---|---|---|---|---|
| Depth (km) | 10 | 13 ± 3 | 10 | 12 ± 2 |
| Lat (°N) | 34.1 | 34.17 ± 0.07 | 34.2 | 34.19 ± 0.08 |
| Lon (°E) | 25.7 | 25.72 ± 0.04 | 25.7 | 25.73 ± 0.05 |
| Strike (°) | 95 | 99 ± 5 | 255 | 249 ± 14 |
| Dip (°) | 50 | 53 ± 5 | 40 | 39 ± 2 |
| Rake (°) | 95 | 106 ± 15 | 75 | 64 ± 11 |
| Slip (m) | 0.50 | 0.68 ± 0.15 | 0.55 | 0.75 ± 0.16 |
| Time.shift (min) | 1 | 1.7 ± 0.7 | 2 | 1.9 ± 0.8 |



Figures 5-7 help to visualise the parameter variability and how the best source models are characterised. The marginal (Figure
5) and the joint distributions (Figure 6 and 7) are provided for the two planes. Marginal and joint distributions provide an
additional measure of the uncertainties. Narrower distributions suggest that the corresponding parameters are better resolved
than those characterised by broader ones.

The strike angle for plane B and the dip angle for plane S show a strongly "preferred" value (diagonals of Figures 6 and 7).
The rake angle does not show a real preferential value: evidently, we do not have enough precision to discriminate at this level
of resolution. Plane B solutions are characterised by a larger depth dispersion and by a higher average depth value. However,
the depth of 20 km almost never occurs, suggesting the occurrence of a shallow event. The slip shows a "bell-shaped"
distribution with a peak at 0.65 m and 0.75 m for B and S plane respectively, and significant occurrences in the range 0.50-
0.90; the best source slip is lower than the average, both for plane S and B. S plane solutions are characterised by a slightly
higher slip than B plane solutions. There is a correlation between the slip and depth values: deeper solutions consistently
feature a larger slip. In this case, a lighter correlation also exists between slip and latitude: events further south have a slightly
greater slip, especially for B solutions. As regards the hypocentre determination, establishing a univocal position is not obvious,
also because the delay adds a trade-off in constraining the hypocentre. Consequently, the Longitude is better constrained than
the Latitude since the latter is more strongly correlated with the arrival time given the relative position of the tide-gauges (both
to the north) with respect to the source. The preferred longitude is 25.7°E, with fewer occurrences a little further east and
almost none further west.



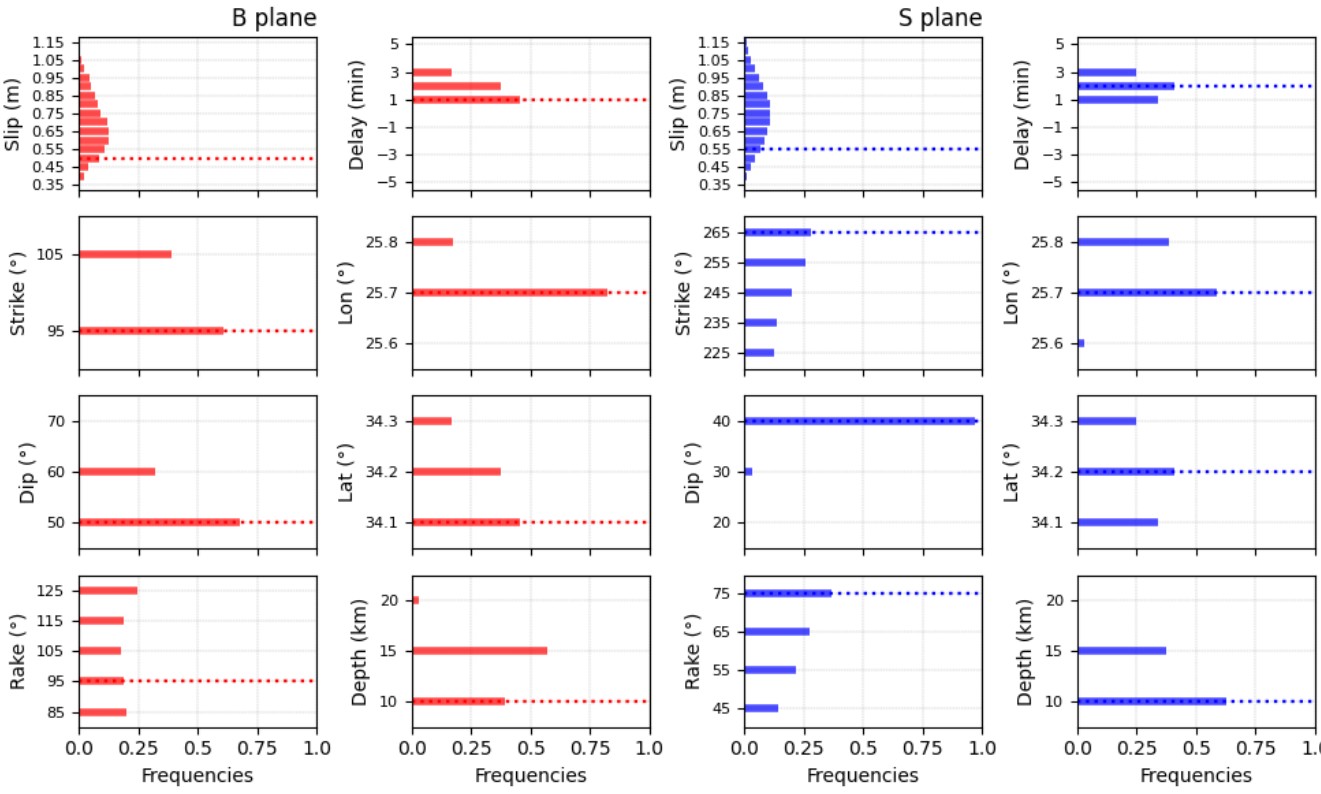

**Figure 5:** Marginal distributions for each of the inverted parameters, considering the first 5 percent of B (1st and 2nd columns) and S (3rd and 4th columns) plane models, those at the left of the red and blue vertical line in Figure 3a. The red and blue horizontal dotted lines mark the best models for the B and S planes, respectively.


**Figure 6:** Joint density distribution for each couple of the back-thrust source's parameters, considering the first 5 percent of B plane models, those at the left of the red vertical line in Figure 3a. The red star identifies the best model.



**Figure 7:** Joint density distribution for each couple of the splay source's parameters, considering the first 5 percent of S plane models, those at the left of the blue vertical line in Figure 3a. The blue star identifies the best model.

The comparison between the observed data and the synthetic ones generated with both the best and the average source models at Ierapetra and Kasos tide-gauge is shown in Figure 8; those corresponding to the two planes B (Figure 8a and e) and S (Figure 8c and g) are plotted separately. Both synthetic signals reproduce quite well the first oscillations. It is interesting to note a possible "clipping" of the negative peak of the signal at ~ minute 27 caused by the insufficient sampling frequency. For what concerns the following peak (minute 28), the average signals result instead to be lower, particularly for the B plane.



In terms of wave fitting, the comparison between the data and the predictions of the average models is only slightly worse than
that found with the best model. The choice between best and average models for both focal solutions is not sufficient for
discriminating, as there are no significant differences, except for the slip, between them. Both can be chosen as the
representative of the best sources' ensembles.


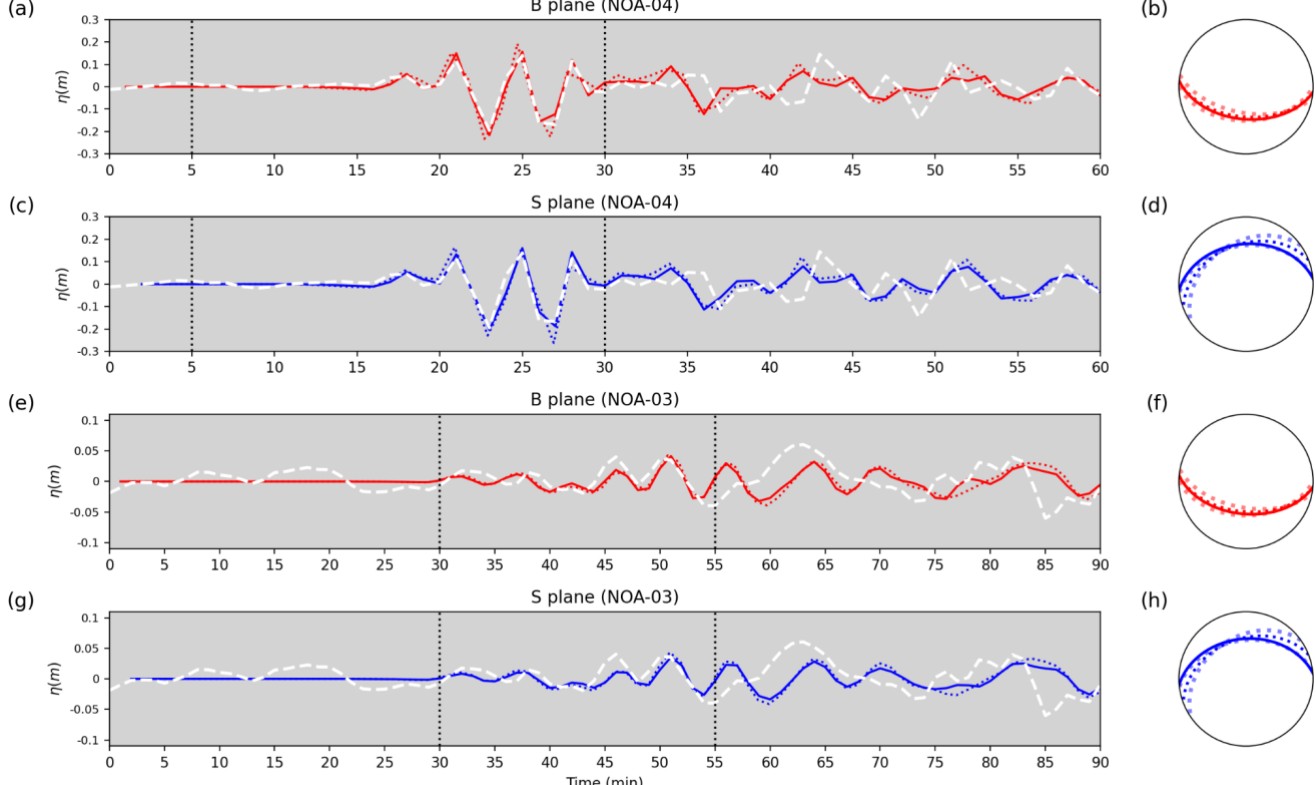


**Figure 8:** Best (solid lines) and average (dotted lines) marigrams obtained at the two stations. Plots (a) and (c) refer to the Ierapetra tide-
gauge (NOA-04) while (e) and (g) to the Kasos one (NOA-03). The white dashed line is the observed water elevation at each tide-gauge. B
plane (in red) refers to the back-thrust solution dipping south; S plane (in blue) refers to the splay fault dipping north. The vertical dotted
lines indicate the limits of the time window used for the inversion. On the right of each marigram plot the stereonets (lower hemisphere)
show the fault orientations corresponding to the best signal (solid line) and the average one (dotted line) with the variability derived from
the standard deviations of Table 2.


The signals belonging to the 5th, 10th, 50th and 100th percentiles of the cost function are shown in Figure 9 (Ierapetra) and
Figure 10 (Kasos) to provide a better idea of what a certain cost function implies in terms of waveform fitting with respect to
the observed data. Significant discrepancies start to appear when including the models in the 10th percentile and beyond,
confirming that all the models with a lower cost function may be equally reasonable solutions.

The synthetic marigrams at Ierapetra and Kasos reproduce quite well the observed tsunami waveforms for the first cycles of
the signal, those carrying most of the source-related information. As discussed above, the agreement worsens as time
progresses due to the possibility of not well-modelled propagation complexity around the tide-gauge. After roughly half an
hour from the tsunami first arrival, there is a larger and larger deviation between the synthetic and the observed marigrams
(Figure 8).

Overall, the results do not conclusively indicate that one focal plane should be preferred over another, and both solutions
remain possible.

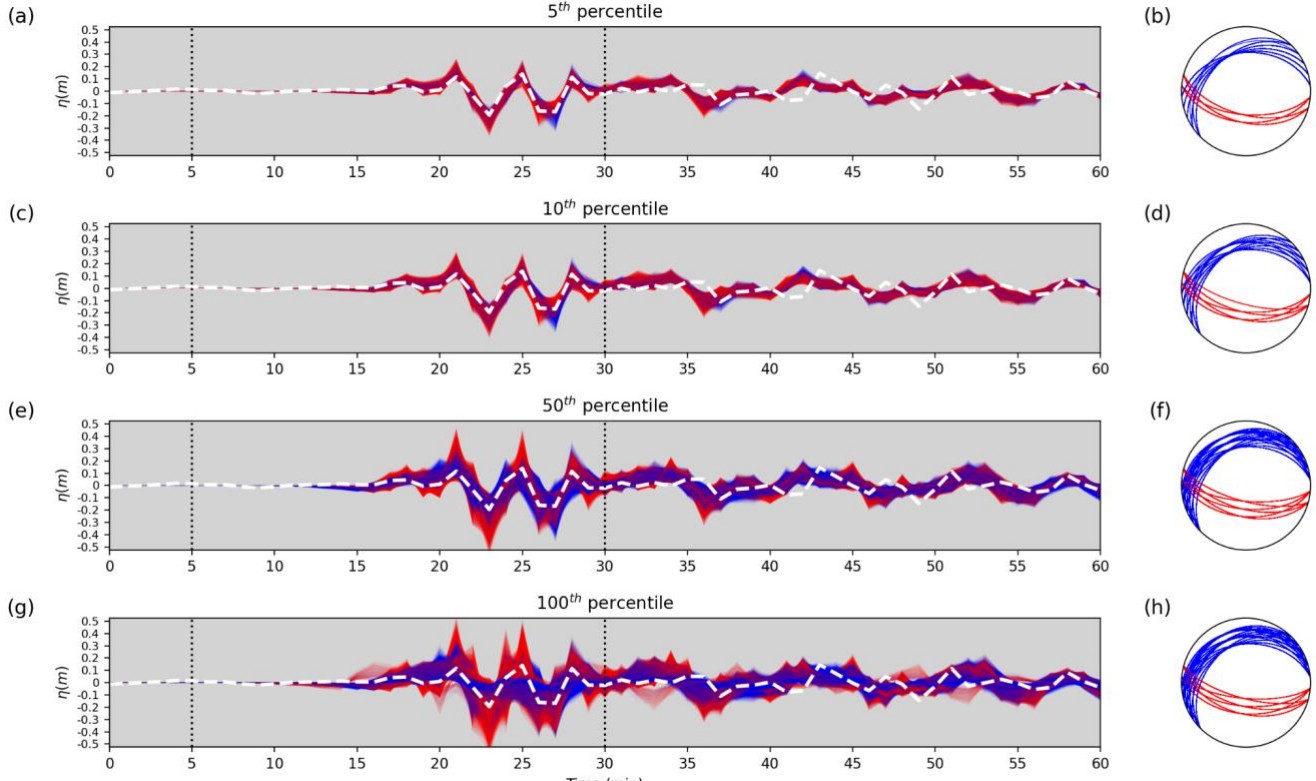


**Figure 9** From top to bottom, the left-hand side panels (a, c, e, g) show the marigrams of the events, ordered by cost function value,
corresponding to the 5th, 10th, 50th, and 100th percentiles. The white dashed line is the observed water elevation at the Ierapetra tide-gauge


(NOA-04). The vertical dotted lines indicate the limits of the time window used for the inversion. The stereonets (lower hemisphere) on the
right-hand side (b, d, f, h) show the fault plane variability corresponding to the synthetic waveforms. Red and blue refer to plane B (back-
thrust solutions) and plane S (splay fault solutions), respectively, both for waveforms and fault planes.

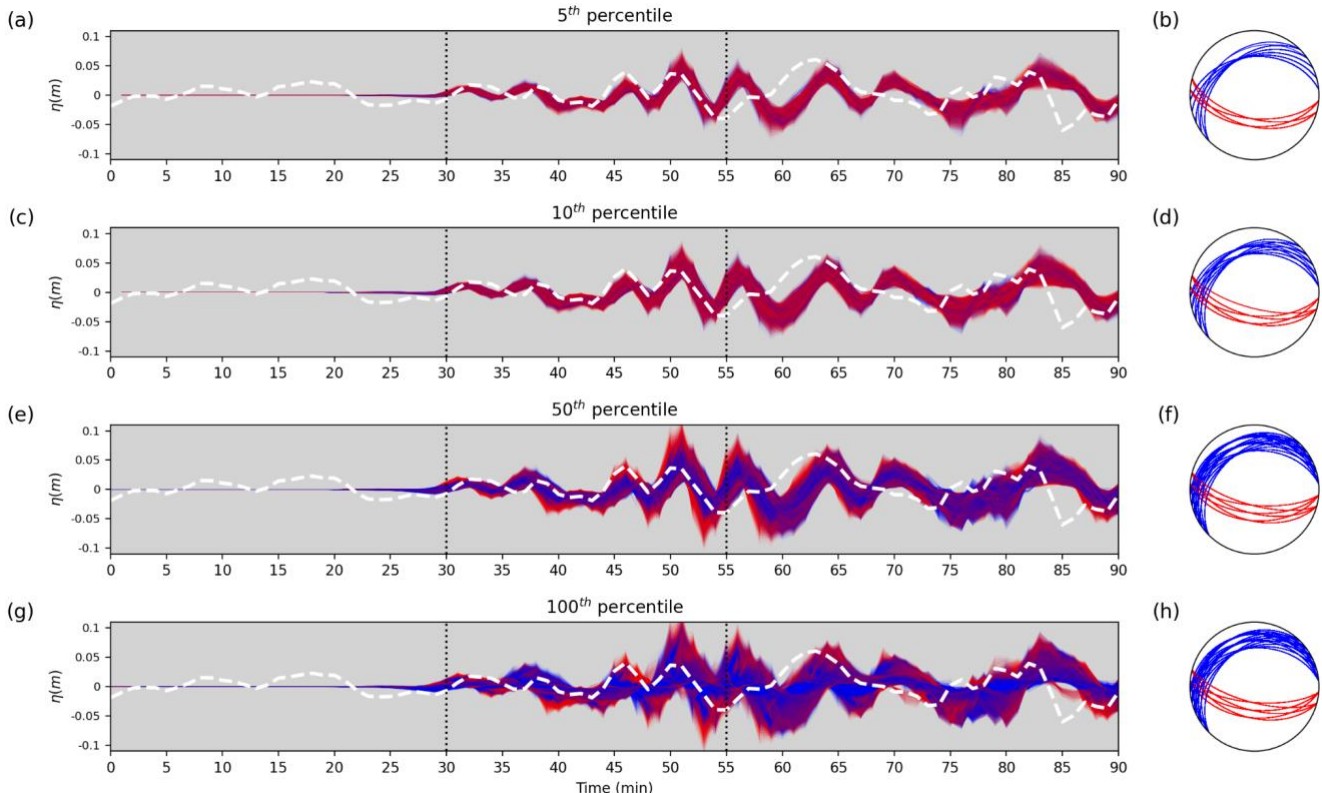


**Figure 10:** The same as Figure 9 but for the Kasos tide-gauge (NOA-03) signal.
**4 Discussion**
We constrained the source model of the 2020 Cretan Passage earthquake (Mw 6.6) by comparing the sea level observations at
two tide-gauges with the synthetic tsunami waveforms.
We could use only one tsunami record not too distant from the source and one farther away. The availability of more
instruments would be precious for both real-time operations and event characterisation. Moreover, a better characterisation of
harbour response and the implementation in the future of high-resolution in-harbour propagation could be important,
particularly considering that deep-sea instruments are nearly absent in the Mediterranean Sea.

We compared the waveforms generated with our solutions with those we simulated using two different source models already
published for the 2020 Cretan Passage tsunami: the one presented by Wang et al. (2020; "W" model hereafter), who use the





event as a test-case for a hypothetical offshore bottom pressure gauges network around Crete Island, to assist tsunami early
warning through real data assimilation, and the Heidarzadeh and Gusman (2021) model ("HG" model hereafter), obtained by
inversion of the same tsunami dataset we used in this study.

Figure 11 displays the marigrams calculated with our preferred models together with the waveforms generated by W and HG
models. The W waveform tends to overestimate the observed signal, both at Ierapetra and Kasos tide-gauge. The HG waveform
reproduces well the observed signal at the Ierapetra station, while it overestimates the signal around minute 50 at Kasos. The
cost functions associated with the four models, evaluated as described in Section 2, are 0.097, 0.104, 0.583, 0.253 for our B
and S planes and for W and HG models, respectively. Using these values, and assuming a rigidity of 33 GPa, consistently with
Leonard (2014)'s scaling relationships, the seismic moment associated with the four source models is 6.63, 7.29, 11.9, 11.1
($\times 10^{18}$) Nm, corresponding to Mw 6.5, 6.6, 6.7, and 6.7, respectively.

The W model, whose waveform presents the largest misfit, consists of a single fault (20 km × 12 km) with a uniform slip of
1.5 m. The epicentre is at 34.205°N, 25.712°E, and the top depth of the fault is 11.5 km; strike, dip, and rake angles are 229°,
31°, and 46°, respectively. These parameters are based on the W-phase focal mechanism solution of the USGS. The slip value
is significantly larger than in our preferred models, and it can explain the overestimation. When the same source is used by
Wang et al. (2020; see their Figure 9), the agreement between the synthetic and observed waveforms is better. However, Wang
et al. (2020) used a bathymetric grid with a resolution of 30 arcsec (~ 925 m), while we used a nested grid approach with a
resolution up to 10 m around the tide-gauge positions (see Section 2). This likely guarantees a better convergence of the
numerical simulation of the relatively short wavelengths characterising this tsunami and explains the difference. When using
a lower resolution, the waveforms can only be reproduced by artificially increasing the fault slip. The role of accurate
bathymetry is of fundamental importance to ensure accurate tsunami simulations also for source characterisation.

The HG model, with assigned location and focal mechanism (reported in the Introduction), presents a source dimension of
40×30 km and a heterogeneous slip distribution with a maximum slip of 0.64 m and an average slip of 0.28 m. In this case,
high-resolution modelling is used around the tide-gauges as well. The slip value of our sources is quite larger than their average,
but associated with a smaller fault (see Table 1). The overall higher cost function value for the HG model retrieved with our
setup can be explained by the fact that the inversion time windows are 13 and 10 minutes for Ierapetra and Kasos tide-gauges,
respectively, much shorter than the ones used in this study (Section 2).







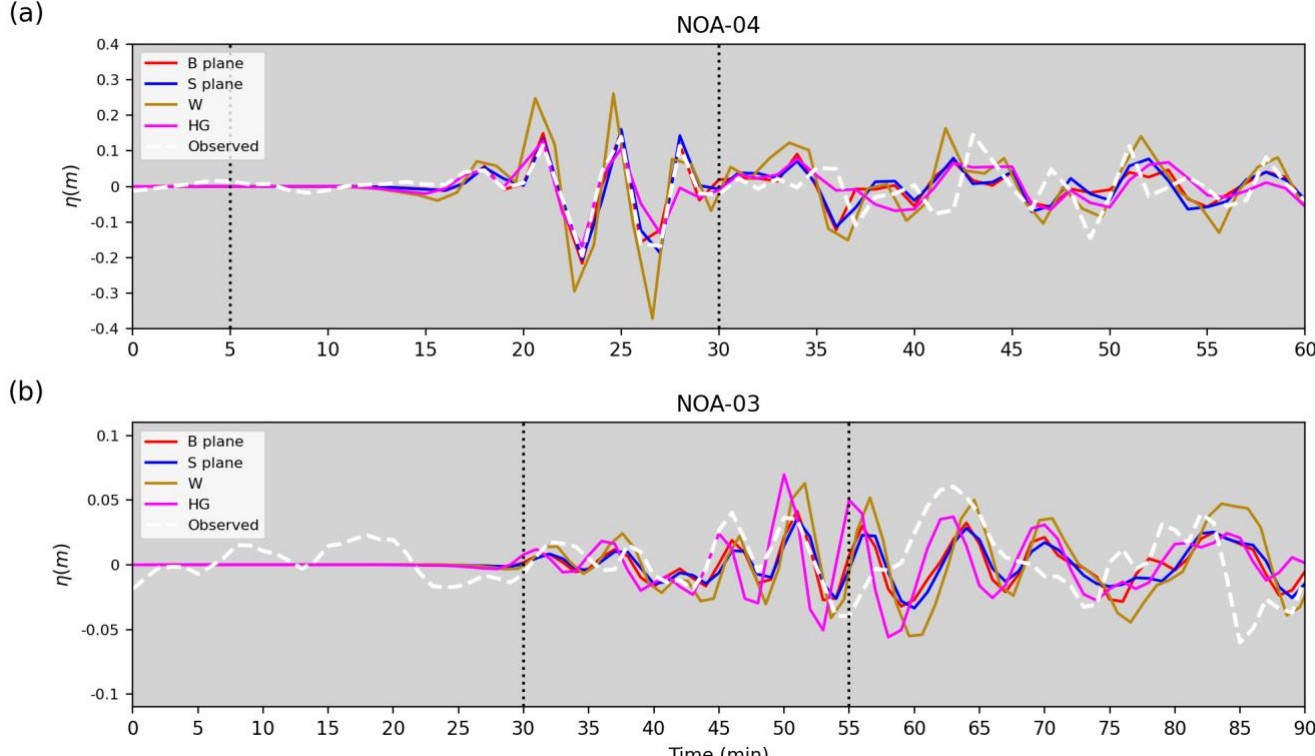


**Figure 11:** Waveforms obtained at the Ierapetra NOA-04 (a) and Kasos NOA-03 (b) tide-gauges by the best source models of the back-thrust solution (the B plane in red), the best of the splay fault solution (the S plane in blue), the fault defined by Wang et al. (2020) and the one by Heidarzadeh and Gusman (2021). The vertical dotted lines indicate the limits of the time window used for the inversion.

Starting from the available focal mechanisms, we explored two thrust faulting solutions (Figure 12), a north-dipping reverse splay fault (plane S) and a south-dipping back-thrust (plane B). We found a slightly better agreement for the waveforms corresponding to the B plane with respect to those of the S plane (Figure 4). However, this difference is not big enough to draw a strong conclusion concerning the causative fault of this earthquake.

Despite this ambiguity between the two fault planes (S and B), still important considerations emerge from this study. Both solutions seem shallow enough to indicate that the earthquake was embedded within the inner parts of the HASZ accretionary wedge, thus excluding either a subduction interface or intraslab earthquake. In particular, the strike of the B plane and the dip of the S plane contribute to excluding a subduction interface earthquake.





From the geological viewpoint, plane B could represent a back-thrust fault accommodating the contraction of the inner parts
of the Mediterranean Ridge against the Cretan backstop. This south-eastern Cretan margin is surrounded by the double Pliny
and Strabo trenches system, which have been related to back-thrust fault activity (Camerlenghi et al., 1992; Leite and Mascle,
1982; Chaumillon and Mascle, 1997). Back-thrusting is considered to be the cause of the formation of a topographic
escarpment separating the wedge from the Inner Ridge backstop (Kopf et al., 2003). The plane S could represent the
reactivation of one of the thrusts marking the advancement of the deformation front within the accretionary wedge above the
main decollement or a splay fault emanating directly from the subduction interface.

In either case, the orientation of the fault plane and the slip direction are compatible with the long-term kinematic indicators.
Within the region of the HASZ where the Cretan Passage earthquake occurred, in fact, the average direction of convergence
is ~ 200-220° from GPS velocity data (Reilinger et al., 2006; Floyd et al., 2010; Noquet, 2012) and the azimuth of the maximum
horizontal stress (SHmax) is 0-20° (Carafa and Barba, 2013). The splay fault S features a small left-lateral slip component,
which is consistent with the increasingly oblique convergence in the eastern branch of the HASZ (Bohnhoff et al., 2005;
Yolsal-Çevikbilen and Taymaz, 2012).

The combination of the shallow depth and the high dip angle plays a key role in determining the tsunamigenic potential
associated with the fault. The steeper dip angle and the shallower depth tend to produce a vertical deformation whose
tsunamigenic potential is more pronounced than that induced by the very low-angle interface earthquakes of similar magnitude.
Note, however, that the dip angle of the two proposed solutions is higher than those derived from seismic reflection profiles
for these types of thrust faults in the region (Kopf et al., 2003).

For example, the moderate earthquake of Mw = 6.45, which occurred on July 1, 2009 (Bocchini et al., 2020), was the cause of
a local tsunami because it ruptured in the overriding crust as for the 2020 Cretan Passage earthquake. Conversely, other larger
earthquakes occurred nearby, apparently without generating a tsunami. Just focusing on the portion of the Hellenic trench
south of Crete, this is, for example, the case of the Ms 7, December 17, 1952, earthquake occurred at a depth of about 25 km
(Papazachos, 1996), and the Ms 6.5, May 4, 1972, earthquake occurred at ~ 40 km depth (Kiratzi and Langston, 1989).





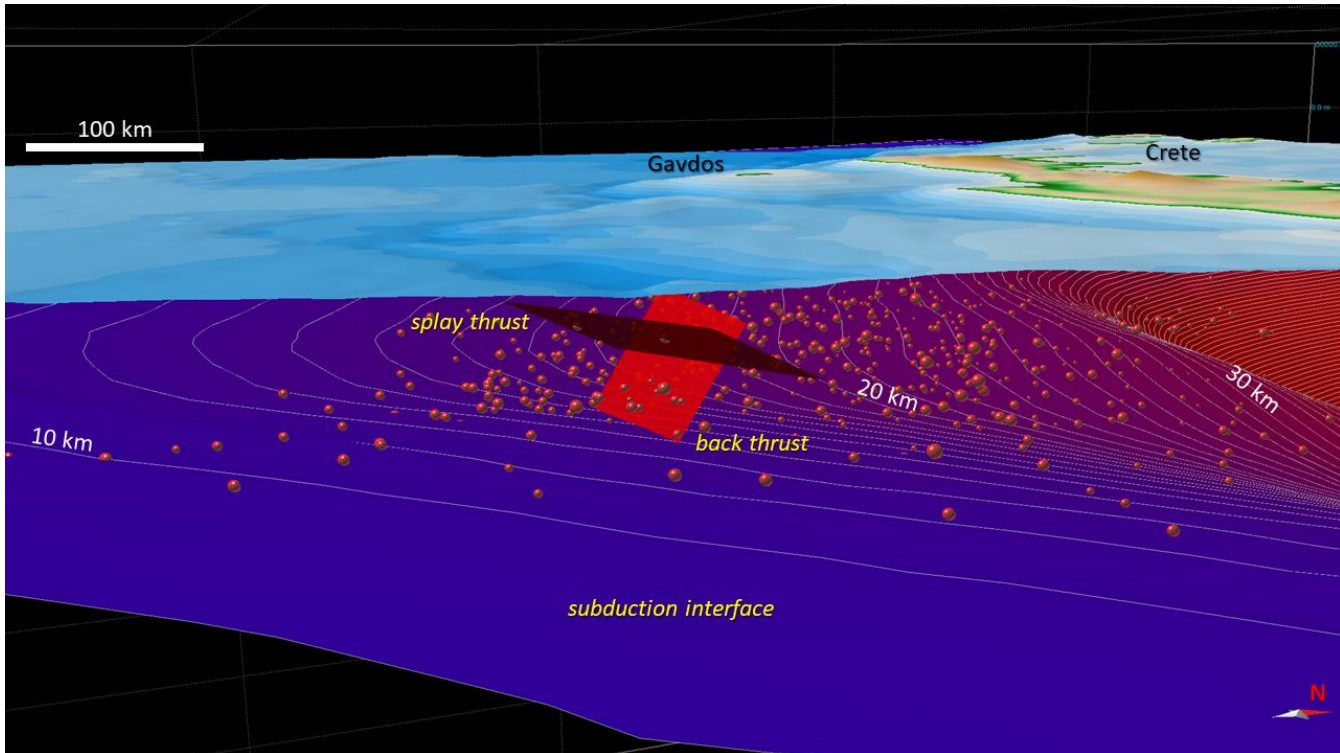

**Figure 12:** Oblique view, looking westward, of the fault planes obtained in this study and their relation with the subduction interface shown by depth contours (white lines) and the aftershock seismicity (red spheres) until 18/04/2021.

## 5 Conclusions

We investigated the seismic fault structure and the rupture characteristics of the Mw 6.6, May 2, 2020, Cretan Passage earthquake through tsunami data inverse modelling. Our results confirm the indication from moment tensor solutions that this was a shallow crustal event with a reverse mechanism within the accretionary wedge rather than on the Hellenic Arc subduction interface.

Using two marigrams, only one of which in the near field with respect to the seismic source, we could highlight important characteristics of this earthquake, especially from a tsunamigenesis perspective, although the adopted method and the limited data available did not prove sufficient to isolate the main focal plane. The sea-level heights recorded at Ierapetra and Kasos tide-gauges identify two possible ruptures: a steeply sloping reverse splay fault and a back-thrust rupture dipping south, with a more prominent dip angle. The a-posteriori appraisal of the ensemble of models tested allows for a slight preference for the south-dipping back-thrust over the splay fault.

Nevertheless, both are high-angle reverse faults in the upper plate above the plate interface with a tsunamigenic potential higher than that of interplate earthquakes of similar or even slightly larger moment magnitude.




This is important for seismic and tsunami hazard assessment, since the presence of shallow crustal ruptures should not be
overlooked in an area where subduction interface (interplate) events are also possible. Note that, for example, the recent
NEAMTHM18 tsunami hazard model considered the possibility of crustal faults rupturing everywhere in the overriding plate
(Basili et al., 2021).

Although the tsunami did not cause damages or victims, the event represents yet another testimony of how such events are
frequent and typical in the Mediterranean and, particularly, along the Hellenic arc. In addition to this, the near-source nature
of the event should be emphasised. Despite the improvements and developments carried out by the NEAMTWS Tsunami
Service Providers in recent years, that have proven to be capable of issuing tsunami messages within 10 minutes after the
earthquake origin time (Amato et al., 2021), the early tsunami arrival (tenths of minutes or less) at the closest coasts leaves
very little time for warning, which is probably the case in many regions in the world. Then, together with an efficient warning
system, education, awareness and preparation remain by far the most cost-effective investments for local tsunamis (Imamura
et al., 2019). The 2020 Cretan Passage earthquake is another reminder of the tsunami risk in the Mediterranean Sea, but also
of the fact that it is extremely appropriate to promptly react to felt shaking, since also moderate earthquakes that are shallow
and occur on steep faults may generate a significant and dangerous tsunami.

*Acknowledgements.* The research reported in this work was supported by OGS and CINECA under HPC-TRES program award
number 2020-01, and co-funded by the Italian flagship project RITMARE. RB acknowledges the resources made available by
the SISMOLAB-3D at INGV.

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
