# Peer review of "Characterisation of fault plane and coseismic slip for the May 2, 2020,"

_Natural Hazards and Earth System Sciences, 2021_

## Author Comment (AC2)

**Supplementary material**

Enrico Baglione[1,2], Stefano Lorito[2], Alessio Piatanesi[2], Fabrizio Romano[2], Roberto Basili[2], Beatriz Brizuela[2], Roberto Tonini[2], Manuela Volpe[2], Hafize Basak Bayraktar[3,2], Alessandro Amato[2]

[1]Istituto Nazionale di Oceanografia e di Geofisica Sperimentale (OGS)- Sgonico (TS) – Italy
[2]Istituto Nazionale di Geofisica e Vulcanologia, Sezione di Roma 1, Via di Vigna Murata 605, 00143, Roma, Italy
[3]Department of Physics "Ettore Pancini", University of Naples Federico II, Naples, 80126, Italy

*Correspondence to*: Enrico Baglione (enrico.baglione@ingv.it)

**1 Introduction Slip linearity assumption**

To test the slip linearity assumption we evaluate some waveforms, derived from the same source parameters, except for the slip value. We considered some different slip values, smaller and larger than unity and then normalized the waveform signals to be associated to the same slip value. The difference in terms of waveform is almost imperceptible and less than 5% of the signal. This is true especially for the first part of the signal, the one that contains the most information about the source.

[Figure]

**Figure S.1** Waveforms obtained at the Ierapetra NOA-04 (a) and Kasos NOA-03 (b) tide gauges by a target source model for different slip values. The different colours associated to the different slip values are summarised in the legend. The tested slip values are the unitary slip, the edges and the middle value of the slip interval considered in the inversion procedure.

**2 Results of the application to the May 2, 2020, Mw 6.6 Cretan Passage earthquake, including the Kasos record**

We reported here the results of the inversion following the same method presented in the main manuscript, but giving to the NOA-03 Kasos station a weight in the inversion procedure.

The overall cost function is a weighted average of two cost functions calculated on the two considered tide-gauges. The weights are chosen such that $\frac{w_{NOA-03}}{w_{NOA-04}} = 0.2$, equivalent to the ratio of the maximum tsunami amplitude registered at the two tide-gauges in the first half an hour after the tsunami arrival. The higher sensitivity of the Ierapetra signal to the source details is not surprising, since the Kasos station is much further away from the source, and the associated recorded marigram shows a very low peak-to-trough excursion and a lower signal-to-noise ratio.

[Figure]

**Figure S.2** (a) Cost function distribution for the back-thrust (red) and the splay (blue) models; the vertical dashed lines indicate the 5th percentiles for each of the two focal solutions. (b) Histogram of the cost function values for all the models considered. The vertical dashed lines represent the 5th, 10th, 50th (median), 90th and 95th percentile.

**Table 2:** see Table 2 of the mail article for details.

|  | Best model plane B | Average model (5th) plane B | Best model plane S | Average model (5th) plane S |
|---|---|---|---|---|
| Depth (km) | 10 | $13 \pm 3$ | 10 | $12 \pm 2$ |
| Lat (°N) | 34.1 | $34.17 \pm 0.07$ | 34.2 | $34.19 \pm 0.08$ |
| Lon (°E) | 25.7 | $25.72 \pm 0.04$ | 25.7 | $25.73 \pm 0.05$ |
| Strike (°) | 95 | $99 \pm 5$ | 255 | $249 \pm 14$ |
| Dip (°) | 50 | $53 \pm 5$ | 40 | $39 \pm 2$ |
| Rake (°) | 95 | $106 \pm 15$ | 75 | $64 \pm 11$ |
| Slip (m) | 0.50 | $0.68 \pm 0.15$ | 0.55 | $0.75 \pm 0.16$ |
| Time.shift (min) | 1 | $1.7 \pm 0.7$ | 2 | $1.9 \pm 0.8$ |

[Figure]

**Figure S.3** Marginal distributions for each of the inverted parameters, considering the first 5 percent of B (1st and 2nd columns) and S (3rd and 4th columns) plane models, those at the left of the red and blue vertical line in Figure 3a. The red and blue horizontal dotted lines mark the best models for the B and S planes, respectively.

[Figure]

**Figure S.4** Joint density distribution for each couple of the back-thrust source's parameters, considering the first 5 percent of B plane models, those at the left of the red vertical line in Figure 3a. The red star identifies the best model.

[Figure]

**Figure S.5** Joint density distribution for each couple of the splay source's parameters, considering the first 5 percent of S plane models, those at the left of the blue vertical line in Figure 3a. The blue star identifies the best model.

[Figure]

**Figure S.6** Best (solid lines) and average (dotted lines) marigrams obtained at the two stations. Plots (a) and (c) refer to the Ierapetra tide gauge (NOA-04) while (e) and (g) to the Kasos one (NOA-03). The white dashed line is the observed water elevation at each tide gauge. B plane (in red) refers to the back-thrust solution dipping south; S plane (in blue) refers to the splay fault dipping north. The vertical dotted lines indicate the limits of the time window used for the inversion. On the right of each marigram plot the stereonets (lower hemisphere) show the fault orientations corresponding to the best signal (solid line) and the average one (dotted line) with the variability derived from the standard deviations of Table 2.

[Figure]

**Figure S.7** From top to bottom, the left-hand side panels (a, c, e, g) show the marigrams of the events, ordered by cost function value, corresponding to the 5th, 10th, 50th, and 100th percentiles. The white dashed line is the observed water elevation at the Ierapetra tide gauge (NOA-04). The vertical dotted lines indicate the limits of the time window used for the inversion. The stereonets (lower hemisphere) on the right-hand side (b, d, f, h) show the fault plane variability corresponding to the synthetic waveforms. Red and blue refer to plane B (back-thrust solutions) and plane S (splay fault solutions), respectively, both for waveforms and fault planes.

[Figure]

**Figure S.8** The same as Figure S.7, but for the Kasos tide-gauge (NOA-03) signal.

---

## Author Response (AR1)

**ANSWERS TO REFEREE 1**

**General comments**

- **Tide gauges do not typically report instantaneous water level measurements. Instead, tide gauges average water level values sampled at a higher rate (than the data output rate). Therefore, downsampling the numerical mareograms every minute (line 190) does not produce an equivalent signal to the tide gauge (averaged) recording. Moreover, the sampling period typically corresponds to the output signal period, thus introducing a time shift of typically half the sampling period. These factors should be taken into account in the inversion and should lead to different results.**

Thank you for the comment, we did not express this concept well in the article. Time series of the tsunami were saved at a sampling rate higher than 1 minute. As the tide gauge data present a sampling of one value per minute, we decided to resample the synthetic signal to the observed data sampling rate. The resampling has been made through a linear interpolation with python.

- **The choice of including the Kasos tide gauge signal in the source inversion or not is not straightforward since the signal to noise ratio is so low. Fig. 8e-g show that particularly for the first 2-3 waves (up to minute 44), the signal to noise ratio is about 1 to 1. Lines 210-215 of the manuscript explain how the Kasos tide gauge signal is assigned a smaller weight, but still its inclusion is questionable when the quality of the recorded data is so poor. Adding random noise with a higher percentage of the clean synthetic waveform amplitude variance to the Kasos tide gauge in the test of Section 2.4 should provide an estimate of how much the low signal to noise ratio of the Kasos tide gauge affects the inversion results.**

This is true. Accepting this suggestion, we modified the inversion procedure by excluding the signal recorded at Kasos tide gauge station. The results do not differ significantly from the previously presented calculation, underlining that this station, located in the far-field, does not significantly constrain the tsunami source for this specific event. However, we used the tsunami forward modeling at Kasos station as an independent verification of the tsunami source estimated through the inversion of the Ierapetra waveform.

**Figure 4, 5, 6, 7, 8, 9** and **Table 2** are updated according to this modification. The changes are neither significant nor appreciable, in particular as regards the waveform signals.
**Figure 10 has been removed,** as Kasos station is no more included in the source inversion.

Kasos signals remain instead in **Figure 7** and **Figure 11** (that now becomes **Figure 10**), as they are just reported as predicted waveforms.

We also added in the supplementary material file the results obtained assigning to the Kasos station a weight that is 1/5 the Ierapetra one.

- **Lines 190-191: The authors state that "We assumed linearity of the slip amount and the tsunami to obtain the scenarios for different slip values". Tide gauges, unlike deep water pressure sensors typically used for linear source inversions, are located in the nearshore where waves are clearly nonlinear. I'm having trouble believing that the linearity assumption is valid without testing it at each (tide gauge) location where it is used. Since the linearity assumption is key to the slip inversion, the authors should include a separate subsection in section 2 where the linearity assumption is verified.**

Thanks for this comment, that is an important point in the methodology; now we verified the slip linearity assumption. Using a set of source parameters, we modelled the tsunami waveform at Ierapetra considering some different slip values, smaller and larger than unity; then we normalized the synthetic waveforms to be associated to the same slip value. The results show that the difference between the waveforms (the target simulated with a unitary slip and the one simulated with a different slip value and renormalised) is less than 5% of the target signal. Now we report the test in the supplementary material: both the unitary slip, the edges of the slip interval adopted in the inversion procedure are considered. We modified the main text (section 2.2) accordingly.

- **The wave period of this particular tsunami is relatively small (2-5 min), and water depth values at the source region reach ~3000-4000 m. Thus, wave energy in the source region is certainly contained in the intermediate (kh~1) water range (outside the kh<π/10 shallow water range). Since a shallow water model was used, frequency dispersion was not considered per se in this study. Early wave arrival of the Green's functions is considered in the form of a time shift together with the inaccuracies of the bathymetry etc, but considering frequency dispersion in the form of a fixed time shift is not equivalent to resolving frequency dispersion through higher order terms in the governing equations. This is an epistemic uncertainty that can be alleviated with the use of a dispersive model, although such an undertaking would be very computationally demanding for such a large number of simulations. A short discussion on the effect of frequency dispersion for this particular (small) tsunami event should at least be included in the Data and Methodology section.**

Thanks for the comment. Dispersion effects are not considered in the shallow water governing equations solved by Tsunami-HySEA code to numerically model the tsunami wave in our study. The tide gauge station (Ierapetra) used

to estimate the tsunami source is located sufficiently near to the source, about 80 km; for such a distance, we assume the effects due to dispersion are negligible. This is consistent with the results presented in a recent study by Sandanbata et al. (2021), now cited in the revised version of our manuscript. They analysed how much dispersion and some additional factors, that reduce the tsunami speed, affect short-period tsunamis with dominant periods below ~1000 s. Their simulation results demonstrated that effects of these additional factors are negligibly small at stations up to ~500 km away from the source, whereas the effects appear as an apparent traveltime delay of ~40 s at a station ~1430 km away from the source. Even if considering a delay of such quantity (and it would not be the case for a near-source station as the one we analyze), it would be smaller than the sampling rate of the data and anyway included in the uncertainty assumed in the estimated delay.

Moreover, although the depth near the hypocentre can be high (about 3 km), it reaches a value of 1 km just after 20 km in the direction of the recording station, and then decreases in the remaining 60 km of propagation.

Heidarzadeh et al. (2021), studying the same tsunamigenic event, also assumed the dispersion effects as negligible, justifying the assumption with the fact that the tsunami wavelength is large enough for this approximation.

Now we modified section 2.2 in the main text accordingly.

- **The source rupture area was fixed and the slip magnitude was varied in the source inversion. Also, the authors did not use the seismic moment as a constraint to try different combinations of rupture length, width and slip magnitude. I believe that was done to save computation time since the slip magnitude was accounted for as a linear perturbation of the Green's functions. The use of scaling laws to compute the fault rupture area and derive the initial conditions for the hydrodynamic simulations does not guarantee an agreement with the tsunami recordings. While in this case the authors produce an excellent agreement with the Ierapetra tide gauge recording, I'm not sure whether other source parameter combinations can produce equally good results. The expected implications in the inversion of using scaling laws to fix the rupture length and width should be briefly discussed after line 129.**

The comment is more than legitimate. We decided for a fixed fault size due to the symmetry of the problem, in terms of source size and source position relative to the recording station, that does not allow to constrain the size of the fault along strike direction. There surely can be other combinations that could fit the data equally well because of this symmetry (as mentioned now in section 3). Due to the lack of such constraints, we imposed the source size (length and width) as fixed parameters. We have decided to use Leonard (2014; LE14) relations because they are derived from seismic moment and suitable for a crustal event. We have also evaluated the fault size using other scaling relations, such as WC94 (Wells and Coppersmith 1994) and TH17 (Thingbaijam et al. 2017). The differences are quite small (11% of Length between LE14 and TH17 and 18% of Width between LE14 and WC94) and even smaller if the source area is considered. We believe that these differences are fully absorbed by the variability of the other

parameters, in particular the slip for moment variations, and the hypocentre position to cover a different coseismic deformation location.

Now, we clarify this point in section 2.1.

**Technical corrections**

**- "tide-gauge" should be written as "tide gauge".**

Done.

**- Line 77: Ebeling et al. (2012) is another reference for the 1948 earthquake and tsunami event:**

**Ebeling, C.W., Okal, E.A., Kalligeris, N. and Synolakis, C.E., 2012. Modern seismological reassessment and tsunami simulation of historical Hellenic Arc earthquakes. Tectonophysics, 530, pp.225-239.**

Added, thank you.

**- Line 119: use "topography-" instead of "topo-...".**

Corrected.

**- Lines 162:163: describe the model governing equations in addition to the numerical scheme.**

The model governing equations are now mentioned in the text (non-linear shallow water equations).

**- Lines 172-173: the nautical charts the authors refer to were produced by the Hellenic Hydrographic Service. The issue date of the nautical charts used should also be mentioned in the text because it is an important piece of information.**

Done.

**- Lines 222-226: this is a difficult concept which I did not fully grasp and I believe needs to be explained/presented better.**

True. It was not explained correctly. Now, even removing the Kasos signal from the inversion, it should be clearer.

**- Line 237: what is the definition of ||aj|| here? The square root of the sum of the squares of all (seven) parameters?**

It is. I added this specification to the main text.

**- Line 270: it was not immediately clear to me what "resolution test results presented in Section 2" refers to. Better refer to them using the title of section 2.4, i.e. synthetic test.**

Done.

**- Lines 338-339: difficult to read. Sentence needs to be rewritten.**

Done.

**- Line 342: "The choice…is not sufficient for discriminating…" needs to be rephrased.**

It was rephrased.

**- Lines 397-398: the moment magnitude values resulting from the inversions can also be presented in Table 2.**

Done.

**- Figure 11: What is the sampling period of the W and HG numerical mareograms plotted here? Also, it is difficult for the reader to distinguish the magenta from the red curves.**

The W and HG signals were also resampled to the observed data sampling rate (1 minute) through the same linear interpolation procedure adopted for our synthetic waveforms.
The magenta line was replaced by a black one.

**ANSWERS TO REFEREE 2**

1. **72 "Guidoboni and Comastri, 1997". Wrong citation, those authors reported on the 1303 tsunami not on the 365 one. Suggested citations for the 365 earthquake and tsunami are, among others, the books by Ambraseys (2009) and Papadopoulos (2011).**

   Thank you, corrected.

2. **75. Papadopoulos et al., 2014. The correct citation is Papadopoulos et al., 2012 (see reference).**

   Thank you, corrected.

3. **139. "a steep south-dipping plane". Please say a few words that may support from geotectonic point of view the possibility of considering such a type of fault in that area.**

   The arguments are reported in the discussion section.

4. **337 "Both synthetic signals reproduce quite well the first oscillations". Please mention how many sec are covered by the first oscillations, up to ~30 sec?**

   About 15 minutes, now it is mentioned in the main text.

5. **381 "not too distant from the source". It is better saying "in the near-field domain"**

   Corrected.

6. **492-493 "leaves very little time for warning". This operationally critical point was examined in details by Papadopoulos et al. (2020) as regards the 2 May 2020 seismic tsunami.**

**Figure 1. Please draw an inset to show the region where the study area is situated.**

   The inset was added to Figure 1.

IN THE FOLLOWING PAGES, THE REVISED MANUSCRIPT IS REPORTED WITH THE MODIFICATIONS HIGHLIGHTED IN YELLOW FOR REFEREE 1 COMMENTS, AND GREEN FOR REFEREE 2 COMMENTS.

[revised manuscript text omitted]